# Selective Determination of Dopamine in Pharmaceuticals and Human Urine Using Carbon Quantum Dots as a Fluorescent Probe

**Xiupei Yang** * , **Fangming Tian, Shaohua Wen, Hua Xu, Lin Zhang and Jie Zeng**

College of Chemistry and Chemical Engineering, Chemical Synthesis and Pollution Control Key Laboratory of Sichuan Province, China West Normal University, Nanchong 637000, China; tfm450756035@163.com (F.T.); shaohuawen@126.com (S.W.); xuhua20210106@163.com (H.X.); zl26274049612021@163.com (L.Z.); zj18291082256@163.com (J.Z.)
* Correspondence: xiupeiyang@cwnu.edu.cn; Tel.: +86-817-256-8081

**Abstract:** A cost-effective and environmentally friendly method was formulated for rapid dopamine (DA) detection that was based on the fluorescence (FL) quenching of carbon quantum dots (C-dots). Upon adding DA to the C-dots' solution, we noticed a regular reduction in their fluorescence intensity. The effects of pH, amount of C-dots, reaction temperature and time on the determination of DA were investigated. Under the optimized experimental conditions, trace amounts of DA could be analyzed. Furthermore, dopamine hydrochloride injection and human urine samples with and without spiked DA were analyzed using the developed sensing system. The procedure was validated following the guidelines of the European Medicines Agency (EMA) in terms of the following: calibration range (0.3–100 μM), linearity ($R^2$ = 0.9991), limit of detection (LOD) (93 nM). Recoveries of dopamine with spiked samples at three different levels were between 95.0 and 105.9%, and the relative standard deviations (RSDs) were within 2.68% (n = 6). This method is simple and suitable for the determination of dopamine in pharmaceuticals and human urine for clinical application. Compared with previous reports, the proposed method offers great advantages including ease of C-dot sensor preparation (one-pot synthesis), environmentally friendly sample preparation by using either water or phosphate buffer solution only, a short response time and selectivity.

**Keywords:** carbon quantum dots; dopamine; fluorescence quenching; yet photochemical probe





## 1. Introduction

As one of the key neurotransmitters, dopamine (4-(2-aminoethyl) benzene-1,2-diol (DA)) plays a critical function in our central nervous system [1]. Neurological disturbances like Huntington's disease, Parkinson's disease and schizophrenia may be caused by low levels of DA [2]. The hydrochloride salt of DA is used clinically as a drug for the therapy of shock, which may be produced by physical injury, heart failure or surgery. For this reason, it is of great clinical significance to detect DA with excellent sensitivity and selectivity in order to investigate and comprehend the mechanisms of neurological disorders.

Some studies have previously reported the detection of DA, such as colorimetric probing [3], electrochemical methods [4], high-performance liquid chromatography (HPLC) [5], LC–MS/MS [6], aptasensor [7], chemiluminescence (CL) [8], resonance rayleigh scattering [9] and other spectroscopic probes [10]. However, some challenges still exist in these current methods that need to be overcome: (i) the prerequisite of expensive analytic instruments; (ii) complicated sample pretreatments; (iii) the cost of organic solvents; (iv) time-consuming testing due to the professional skills of the operator and procedure. For these, it is unfavorable to achieve the request of quick on-site screening. Therefore, developing a rapid, simple, convenient, eco-friendly and sensitive method for the determination of DA has become incredibly valuable and attractive.

Carbon quantum dots (C-dots) are currently considered to be a promising material due to their low toxicity, favorable biocompatibility, good water solubility and excellent photostability, and represent a novel category of fluorescent nanomaterials with a scale of <10 nm. Over the last several years, most studies have concentrated on the characteristics and syntheses of C-dots, but there has been little focus on their applications apart from those in visual detection [11], temperature sensing [12], bioimaging [13], photocatalytic reaction [14], electrocatalytic oxidation [15], molecular imprinting [16] and light-emitting devices [17]. Based on the fluorescence (FL) properties of C-dots, some efforts have been taken in analytical applications of C-dots [18]. In addition, nitrogen- and iron-containing C-dots have been synthesized and successfully utilized in the fluorometric determination of dopamine [19]. Recently, we proposed a simple strategy for the synthesis of fluorescent C-dots through a hydrothermal method using plant material as a carbon source. By employing the new C-dots, we successfully developed several effective fluorescence C-dots for the determination of tartrazine, metronidazole and glutathione, respectively [20–23].

In this research, an environmentally friendly and convenient approach was developed to synthesize fluorescent C-dots via L-arginine hydrothermal treatment, and their analytical application is examined. In view of the fluorescence quenching, an effectual probe proposed by C-dots is suggested for the selective and sensitive analysis of DA. Further, the concentrations of DA in different clinical samples such as human urine and dopamine hydrochloride injection are measured successfully based on the synthesized C-dots.

## 2. Materials and Methods

### 2.1. Chemical Reagents

Dopamine hydrochloride ($\geq$98%), L-arginine (Arg, $\geq$99%), L-cysteine (Cys, $\geq$99%), L-histidine (His, $\geq$99%), L-lysine (Lys, $\geq$98.0%), L-tyrosine (Tyr, $\geq$99.0%), L-serine (Ser), $\geq$99.5%), L-homocysteine (Hcy, $\geq$98.0%), glycine (Gly, $\geq$98.5%), rhodamine 6G ($\geq$99.0%), tartrazine ($\geq$99.0%) metronidazole ($\geq$99.8%), glutathione (98%) and 5-hydroxytryptamine (5-HT, $\geq$98%) were purchased from Aladdin Chemistry Co. Ltd. (Shanghai, China). Disodium hydrogen phosphate dodecahydrate ($Na_2HPO_4 \cdot 12H_2O$, 99.0%), sodium dihydrogen phosphate ($NaH_2PO_4 \cdot H_2O$, 99.0%) and sodium oxalate (99.0%) were obtained from Tianjin Fuchen Chemical Reagents Co., Ltd. (Tianjin, China). Glucose, lactose, ascorbic acid, epinephrine and urea were $\geq$98% and from Maoye Chemical Reagent Co. Ltd. (Chongqing, China). Other reagents, such as KCl, $MgSO_4 \cdot 7H_2O$, $Na_2CO_3$, $CaCl_2 \cdot 2H_2O$, $MnSO_4 \cdot 2H_2O$, $ZnCl_2$, $FeCl_3 \cdot 6H_2O$, $FeCl_2 \cdot 4H_2O$ and $CuCl_2 \cdot 2H_2O$ used in this work were of analytical grade and purchased from Kelong Chemical Reagent Factory (Chengdu, China). Dilute dopamine hydrochloride injection (10 mg/L, Yuanda Pharmaceutical China Co., Ltd., Wuhan, China) and human urine were collected from the school hospital. All solutions were prepared using ultrapure water.

### 2.2. Apparatus

The absorption spectrum of C-dots were obtained using a Shimadzu UV-2550 UV-vis absorption spectrophotometer (Kyoto, Japan). Fluorescence tests were conducted with a Cary Eclipse fluorescence spectrophotometer (Varian, Palo Alto, CA, USA) equipped with a 1.0-cm quartz cell. The samples' infrared spectra were accessed on a Nicolet 6700 Fourier transform infrared (FTIR) spectrometer (Thermo Electron Corporation, Waltham, MA, USA) with KBr pellets at room temperature. The element composition was carried out with X-ray photo-electronic spectroscopy (XPS, ESCALAB 250Xi, Thermo Fisher Scientific, Waltham, MA, USA) employing a monochromated Al $K_\alpha$ X-ray source ($hv$ = 1486.6 eV). The XPS results were accumulated at 150 eV constant pass power of the base and analyzer pressure in $10^{-9}$ mbar's analysis chamber. The binding power level was calibrated by evaluating the C1s peak (284.6 eV). A transmission electron microscope (TEM) was used with an acceleration voltage of 200 V on JEM-1200EX (JEOL, Tokyo, Japan).

*2.3. Preparation of Fluorescent C-Dots*

Carbon quantum dots (C-dots) with fluorescence were synthesized via hydrothermal processing based on previous work [20] with some minor modifications. Briefly, the mixture, which contained 0.10 g of L-arginine and 20 mL of water was transmitted into a 50-mL Teflon-lined high-pressure reactor then heated at 180 °C for 6 h. The reactor was then naturally cooled in a ventilated cabinet on an asbestos board. Thereafter, the solution obtained with light yellow was divided into two parts. One of the solutions was filtered directly with a 0.22-μm millipore membrane, while the other solution was filtered with the same membrane after being dialyzed in membrane tubing (molecular weight cutoff ~500) against fresh water for 48 h. Finally, the filtrates of the C-dots as-obtained were placed at 4 °C for use and characterization.

*2.4. Preparation of Samples*

Dopamine hydrochloride injection (10 mg/mL) and human urine samples were tested in this study. Overnight urine samples were obtained from two healthy volunteers, and six parallel samples were tested. The 1.0-mL injection was diluted to 250 mL with ultra-pure water, while the urine sample was appropriately diluted with phosphate buffer solution (PBS, pH = 8.0). All diluted samples were filtered through a 0.45-μm millipore membrane, and the filtrates were collected for further use.

*2.5. Detection of DA*

The detection procedure of DA was achieved in 30-mM PBS (pH = 8.0) at 45 °C. To a 5.0-mL centrifuge tube was added 800 μL of C-dots solution, 1000 μL of PBS and different amounts of DA stock solution (10 mM) or other interfering substances. The last mixture was then diluted to 4.0 mL with water. After maintaining at 45 °C for 1 h, samples' fluorescence spectra were obtained under an excitation wavelength of 295 nm and the slit widths were set at 10/10 nm, respectively. All recovery results were worked out according to the equation

$$Recovery = (C_{measured} - C_{initial})/C_{adde} \tag{1}$$

where $C_{measured}$ is the obtained DA concentration after spiking, $C_{initial}$ is the obtained DA concentration before spiking and $C_{added}$ is the spiking DA concentration, respectively.

## 3. Results

*3.1. Optimization of the Preparation Conditions*

To guarantee the synthesized C-dots' performance, the preparation conditions have been optimized, and the results are presented in Figures S1–S3. As depicted in Figure S1, the fluorescence intensity of the obtained C-dots solution reached a maximum with 20 mL water when using the same amount of L-arginine as a raw material. This may be owing to a certain amount of hydroxyl and carboxyl groups on the surface of the carbon dots. On the one hand, when the amount of solvent is small, the concentration of the carbon dot solution is large, and the carbon dots may agglomerate for hydrogen bonding or electrostatic interaction between the surface groups, thereby causing a slight decrease in fluorescence. On the other hand, when the amount of solvent is large, the concentration of the carbon dot solution is small, which will also weaken the fluorescence intensity.

The fluorescence intensity rose with reaction temperature's rise from 160 to 180 °C but decreased when the temperature exceeded 180 °C in Figure S2. This is because when the reaction temperature is too low, arginine carbonization is incomplete, and the yield of C-dots is low, resulting in a slight decrease in fluorescence intensity. If the reaction temperature is too high, it will lead to excessive carbonization of arginine into small particles to promote agglomeration of the carbon dots product, which also leads to a decrease in fluorescence intensity. As can be observed in Figure S3, the fluorescence intensity rose gradually with the response time up to 6 h, then reduced after 6 h. Based on those results from Figures S1–S3, 20 mL water and 180 °C for 6 h were the optimal conditions for the preparation of C-dots.

As we all know, it is very important to know the fluorescence response of different batches of synthesized nanoparticles. Under the optimized synthesis conditions above, we repeated 6 batches of experiments, and the relative standard deviation (RSD) of the fluorescence intensity of the C-dots solution filtered directly with a 0.22-μm membrane was less than 3%. This result indicates that the fluorescence intensity of the material synthesized by this method was very stable, which is advantageous to establishing a fluorescent probe. In addition, the as-prepared C-dots were further purified through dialysis (molecular weight cutoff ~500) to remove the extra reagent and small molecules. Unfortunately, it was found that the fluorescence intensity of the dialyzed C-dots solution was lower than the non-dialysis C-dot in our experiments. The reason for this phenomenon may be that the fluorescence of the product solution not only derived from the C-dots, but also from the highly fluorescent low molecular weight compounds contained in the product. When these low molecular chemicals are removed by dialysis, their fluorescence intensity decreases. In addition, the highly fluorescent low molecular weight compounds are also subject to a fluorescence quenching upon reaction with distinct analytes. Considering that the sensitivity of a dopamine detection method based on fluorescence quenching should theoretically correlate with the fluorescence intensity of C-dots, we elected to use the C-dots without dialysis for the following experiments.

### 3.2. Characterization of C-Dots

Fluorescence spectrum, UV-vis spectrum, TEM, FTIR and XPS were used to characterize the structure and property of synthesizing C-dots. The UV-vis absorption (a) and fluorescence emission (b) spectra of C-dots are presented in Figure 1. In the spectrum, a peak around 300 nm is attributable to the π–π* transition of C=C [24]. When excited at 295 nm, an optimal emission peak was shown in the photoluminescent (PL) spectrum at about 345 nm. Using Rh 6G as a reference, the quantum yield for the C-dots as-prepared was calculated to be 18.9%, which is two times that of C-dots [20], which may be ascribed to the structure of amide engendered from L-arginine. Similar values of the fluorescence intensity were observed between freshly prepared C-dots and three-months-aged ones. It is provable that the obtained C-dots were extremely stable in the aqueous solution. Although the maximum emission wavelength of the carbon dots is very low (in the UV) and not typical, based on fluorescence quenching by dopamine, they can be used as fluorescent probes for detecting dopamine. The TEM image of the as-synthesized C-dots are shown in Figure 2. It reveals that the C-dots are spherical and non-uniform in particle size, with an average particle size of about 9 nm, and they are well dispersed in the solution.

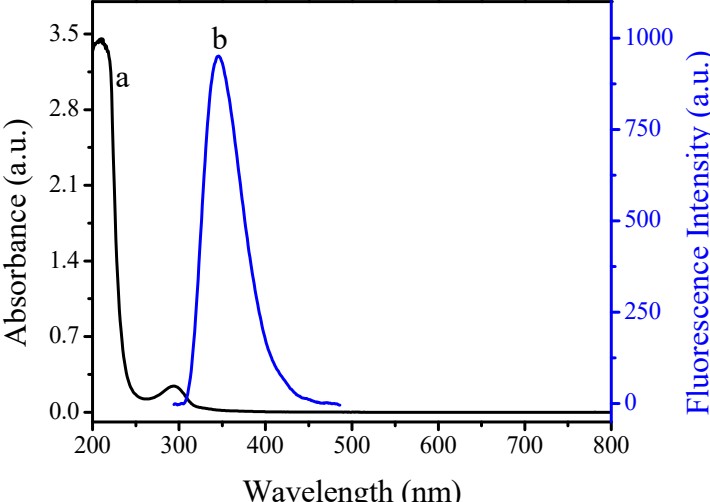

**Figure 1.** UV-vis absorption spectrum and fluorescence (FL) emission spectrum of the carbon quantum dots (C-dots) excited at 295 nm.

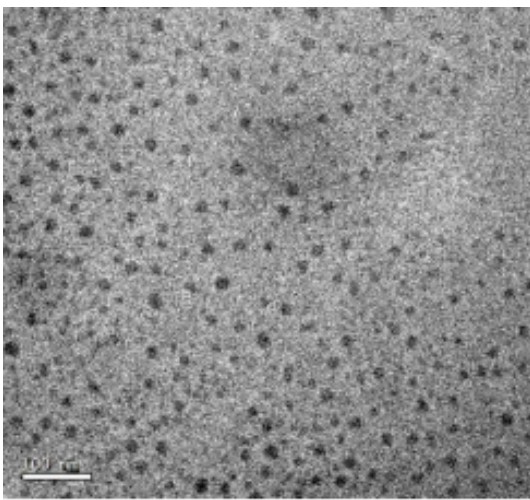

**Figure 2.** TEM image of C-dots.

FTIR was used to analyze and identify the surface of the as-prepared C-dots. It is observable in Figure 3 that the distinctive absorption groups of the −OH stretched the vibration mode at about 3420 and 1091 cm$^{-1}$ and the C−H stretched the mode at 2950 and 1390 cm$^{-1}$, respectively. Besides, the peak appearing at 1650 cm$^{-1}$ might belong to the stretching vibration of C=O/C=C [25]. Therefore, the proof that both the hydroxyl and carboxylic groups were offered through carbohydrates in the L-arginine is supplied by FTIR results. In addition, the broadening of peaks at 3500~3300 cm$^{-1}$ and 1650~1550 cm$^{-1}$ indicate the existence of N-H stretching vibration peaks. The reason for the insignificant peak shape may be affected by the carboxyl vibration peak.

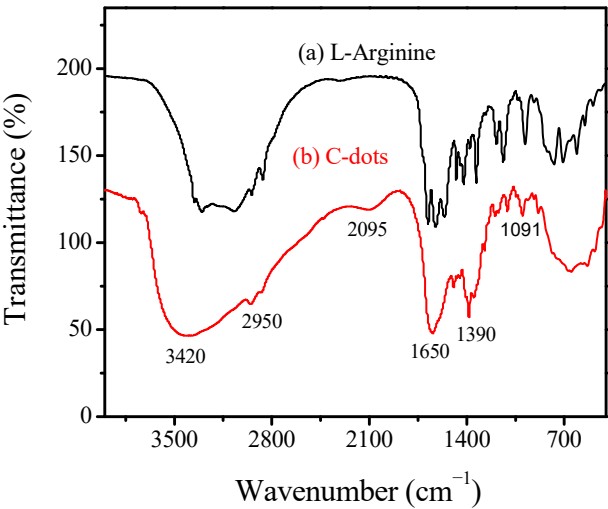

**Figure 3.** The FT-IR spectra of C-dots and L-arginine.

XPS was carried out to further research the structure, components and surface groups of the as-prepared C-dots. The three peaks shown in the nanoparticles' XPS spectrum (Figure 4A) are attributable to C1s, N1s, and O1s, respectively. This suggests that these nanoparticles are principally composed of C and O, as well as a restricted quantity of the N element. The C1s spectrum (Figure 4B) shows three energy peaks of 284.6, 285.3 and 288.2 eV, which might be assigned to C-C/C=C, C-N and C=O, respectively [26]. Two peaks at 398.9 and 399.9 eV in N1s spectrum (Figure 4C) correspond to the C-N-C and C-N groups, respectively. Two peaks shown by the O1s spectrum in Figure 4D at 530.8 and 531.6 eV can be ascribed to the C=O bands and C-O, respectively [27]. The results from the XPS are consistent with those from FTIR.

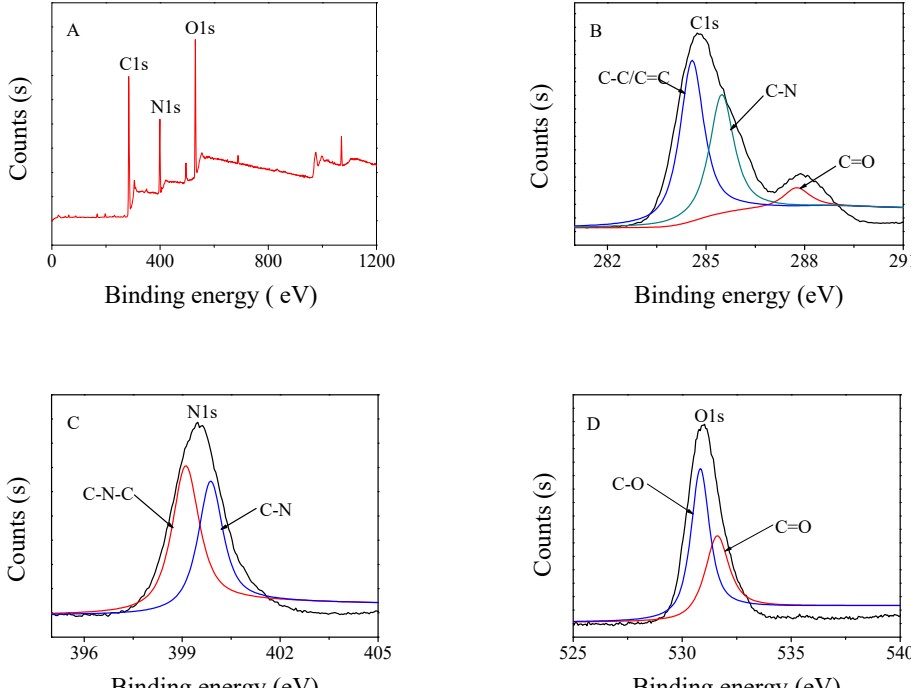

**Figure 4.** Survey XPS spectra of C-dots (**A**) and high-resolution XPS data of C1s (**B**), N1s (**C**) and O1s (**D**) of C-dots.

### 3.3. The Principle of the Fluorescence Probe

Underneath the identical experimental statuses, the fluorescence intensity of the prepared C-dots was decreased significantly with the addition of DA. In view of this phenomenon, we speculate that this fluorescence quenching can be used to create a simple fluorescent probe for the determination of DA. Scheme 1 shows the C-point synthesis strategy and the principle of the DA probe.

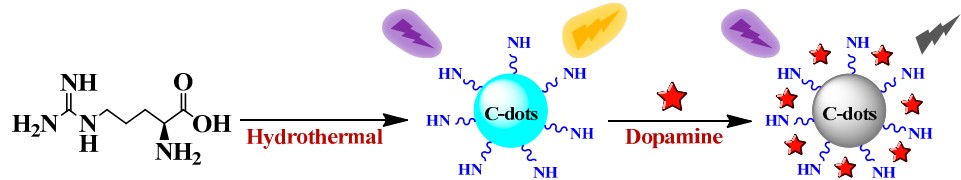

**Scheme 1.** The synthetic strategy for C-dots and the principle of DA sensing.

### 3.4. Mechanism of Fluorescence Quenching

Figure 5 depicts the fluorescence and absorption spectrum of C-dots and DA, where an overlap exists between the absorption spectrum DA and the excitation and emission spectra of C-dots. To avoid the influence of the strong absorption of DA, we corrected the observed fluorescence intensity value [28], and the correction equation is described as follows:

$$F_{corr} = F_{obs} \times 10^{\wedge(\frac{A_{ex}+A_{em}}{2})} \tag{2}$$

where $F_{corr}$ and $F_{obs}$ are the corrected and observed fluorescence intensities, respectively; and $A_{ex}$ and $A_{em}$ are the absorbance of the different concentrations of DA at the excitation and emission wavelengths, respectively.

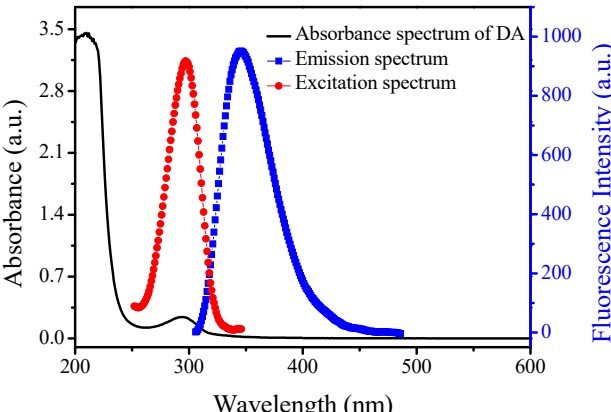

**Figure 5.** UV-vis absorption spectrum of DA, and emission and excitation spectra of the C-dots.

In general, fluorescence quenching includes dynamic quenching and static quenching, even a mixture of the two. Static quenching is a quenching process caused by the complex reaction of quencher and fluorescent molecules in the ground state. In order to explore the quenching mechanism of DA on C-dots, the fluorescence intensities of the C-dots were compared, and the plots were drawn at 25, 35, 45 and 55 °C, respectively. Figure 6 shows that the $F_0/F$ of C-dots is dependent on the concentration of DA at different temperatures, and it conforms to the Stern–Volmer equation in its dynamic quenching effect as follows:

$$F_0/F = 1 + K_q\tau_0[Q] = 1 + K_{SV}[Q] \tag{3}$$

where $F_0$ and $F$ are the fluorescence intensities of C-dots at 345 nm in the absence and presence of DA; however, $F$ was replaced by $F_{corr}$ in this work; $K_{SV}$ and $K_q$ are the Stern−Volmer quenching constant and the biolecular quenching rate constant, respectively; $[Q]$ is the concentration of DA; and $\tau_0$ is the average lifetime of C-dots in the absence of DA. We initially speculated that there was a dynamic quenching effect in the quenching mechanism of DA on C-dots. In addition, it can be seen from Figure 6 that the $F_0/F_{corr}$ value had good linearity at low concentrations of DA, but the linearity changed when the DA concentration increased, with the $F_0/F_{corr}$ deviating from the Y axis to a greater degree. Therefore, we speculate that there is more than one quenching effect on the quenching mechanism of DA [29]. What is more, in Figure S4, the relationship between the C-dot fluorescence ratio $F_0/F_{corr}$ and DA concentration is not a straight line, which also indicates that there may be a mixture of the two mechanisms in the system.

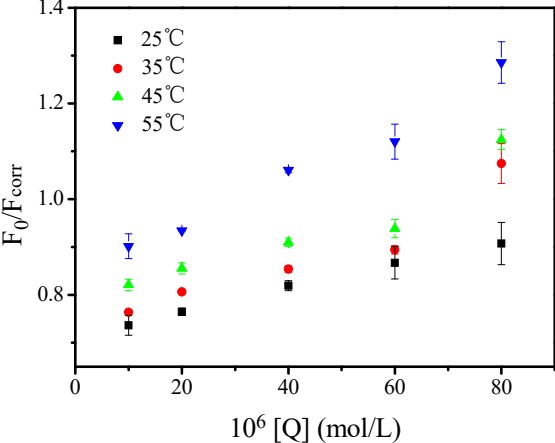

**Figure 6.** Stern−Volmer plots for the system of C-dots−dopamine at 25, 35, 45, and 55 °C, respectively. $F_0$ and $F_{corr}$ are the fluorescence intensity of C-dots in the absence and presence of dopamine (DA), where $F_{corr}$ is the corrected fluorescence value. Conditions: C-dots, 800 μL; PB, 30 mM, pH 8.0.

To further verify that there may be a mixing mechanism, we characterized the fluorescence lifetime of C-dots in the presence or absence DA, and the results are shown in the Figure 7. The fluorescence lifetime value shows that after DA was added, the lifetime value of C-dots was slightly reduced, and the stable lifetime value indicates that there was a static quenching effect under the condition of a high concentration of DA. In addition, there are weak absorption peaks of new substances in the ultraviolet absorption spectrum (Figure S5), and the peak intensity increases with the increase in DA concentration, which may have had an internal filtering effect. Therefore, we believe that the quenching mechanism of DA on C-dots is a mixed form of dynamic quenching and static quenching.

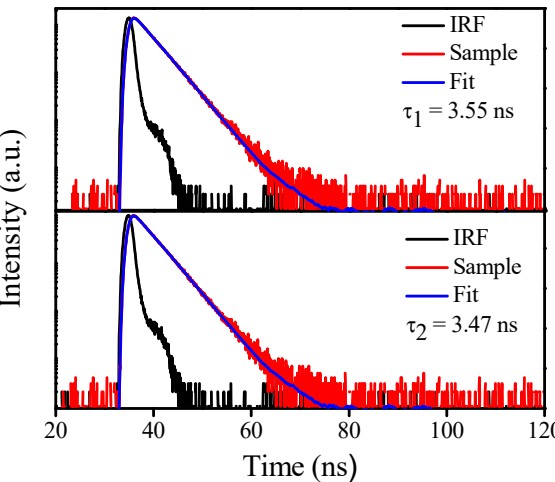

**Figure 7.** The fluorescence lifetime of C-dots in the presence/absence of DA.

### 3.5. Optimal Conditions for DA Detection

#### 3.5.1. Effect of pH

To obtain the optimal pH condition for fluorescence quenching, the quenching efficiency under pH values of PBS at 1.0~13 was investigated when controlling the reaction temperature, time and amount of C-dots as 45 °C, 0.5 h and 800 µL, respectively. The dependence of C-dot fluorescence upon pH in the presence of DA is shown in Figure 8A. In this work, the efficiency of fluorescence quenching is expressed by $F_0/F_{corr}$, where $F_{corr}$ and $F_0$ are the fluorescence intensity of the C-dot solution after and before the addition of DA at 345 nm, respectively. The results show that the C-dots' fluorescence quenching efficiency was poor under both strong and medium acidic conditions (pH = 1.0~6.0). The fluorescence quenching efficiency of the system increased with a pH from 7.0 to 8.0, followed by a gradual decrease with a further increase in pH from 9.0 to 13.0. Why did the fluorescence quenching efficiency of C-dots decrease in both acidic and basic medium? In the case of the low pH, the concentration of oxidized DA was very small, and only edge quenching occurred. With the increase of pH, the proportion of DA oxidized to a quinone structure under oxygen conditions greatly increased. The generation of electron acceptor near C-dots provides a favorable channel for the fluorescence quenching of C-dots. When the pH is too high, the rapid deprotonation of DA will exceed its oxidation, which will weaken the fluorescence quenching of C-dots [30]. The results show that the fluorescence intensity of the synthesized C-Dots was largely affected by the pH value of the medium. This result is consistent with the results of the C-dots modified with hydroxyl and carboxylic/carbonyl moieties [31]. We ultimately chose 8.0 as the optimal pH value for our follow-up study.

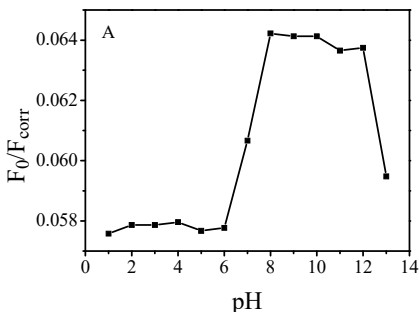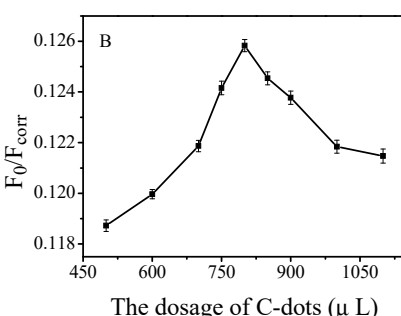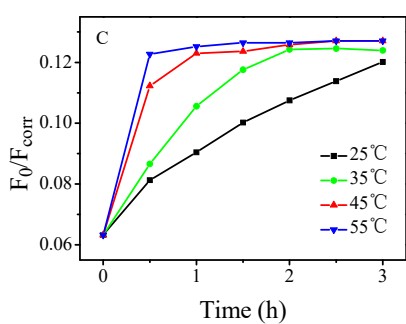

**Figure 8.** Effects of pH of phosphate buffer solution (PBS) (**A**), amount of C-dots (**B**), synthetic time and temperature (**C**) on fluorescence quenching efficiency of the C-dots-DA system. $F_0$ and $F_{corr}$ are the fluorescence intensity of C-dots in the absence and presence of DA, respectively. Conditions: PB, 30 mM; DA, 37.5 µM.

### 3.5.2. Effect of the Amount of C-Dots

The amount of fluorescent material usually has a large influence on the fluorescence quenching experiment. Under the conditions of a 8.0 pH, 45 °C temperature and 1.0 h reaction time, we investigated the condition of the use of C-dots. Figure 8B shows the influence that the amount C-dots had on the efficiency of fluorescence quenching. The highest fluorescence quenching efficiency was achieved when the amount of C-dot solution was 800 µL. Too low or too high dosages of C-dots were not conducive to improvement of the efficiency of fluorescence quenching. Therefore, the use of a 800-µL C-dots solution was used for the detection.

### 3.5.3. Effects of Reaction Temperature and Reaction Time

The influence of reaction time on the fluorescence quenching efficiency of the system at 25, 35, 45 and 55 °C were studied under the conditions of an 8.0 pH and 800-µL C-dots, respectively. As shown in Figure 8C, the fluorescence quenching efficiency of the system was increased at the initial stage of adding dopamine, with the reaction temperature changing from 25 to 45 °C. However, when the temperature was controlled at 25 and 35 °C, the fluorescence quenching reaction proceeded so slowly that the fluorescence intensity of the system was not stable after 2.5 h. Conversely, if the temperature was controlled at 45 and 55 °C, the quenching reaction proceeded relatively quick, and the fluorescence quenching efficiency could reach its maximum within 1.0 h. Considering that the temperature was too low, and the time required for the quenching reaction was too long to construct the fluorescent probe, the results are unfavorable. In addition, the fluorescence intensity tended to be stable and the degree of fluorescence quenching was comparable after 1 h at 45 and 55 °C when we chose a reaction temperature of 45 °C and a reaction time of 1.0 h as the optimal conditions, respectively.

### 3.6. Method Validation

The method was fully verified by the guidelines for a bioanalytical method issued by the European Medicines Agency (EMA). Parameters such as selectivity, calibration curve, linearity range, accuracy, precision, dilution integrity, stability, limit of detection (LOD) and robustness were measured, respectively. In addition, the limit of quantification (LOQ) of this work was also calculated. The whole validation was performed using spiked urine samples from healthy volunteers who were not taking any medication, unless specified.

### 3.6.1. Calibration Range

From Figure 9, we can clearly see that $F_0/F$ and different concentrations of DA have a linear relationship under the same experimental conditions, indicating the dependence of $F_0/F$ on different concentrations of DA. With the increase in the concentration of DA, the fluorescence quenching efficiency of C-dots gradually decreased. As shown in the inset of Figure 9, a linear response can be seen between the decrease in fluorescence quenching

efficiency when the DA concentration increased from 0.30 to 50 μM. The resulting regression curve could be represented as $F_0/F_{corr} = 0.0204c + 1.0153$ ($R^2 = 0.9991$), where $c$ is the concentration of DA (μM) and $R$ is the correlation coefficient. The relative standard deviation (RSD) was 1.64% for the 12 parallel measurements of 10.0 μM DA (n = 12), suggesting that the sensor had excellent precision. It was estimated that the limit of detection (LOD) would be 93 nM (the definition of LOD was $3\sigma/K$, where σ was the standard deviation and K was the slope of the regression curve).

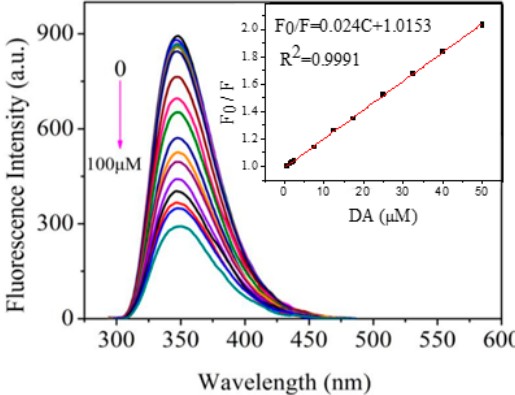

**Figure 9.** The quenching fluorescence spectra of the system by DA. Inset: calibration graph of the fluorescence quenching and different concentrations of DA.

### 3.6.2. Accuracy and Precision

To evaluate the suggested technique's expected applicability, the proposed approach was employed to the determination of DA in human urine and dopamine hydrochloride injection samples. DA in spiked and non-spiked samples was determined to investigate the method's accuracy (recovery). The samples were established by using the proposed technique that was depicted in the experimental portion to eradicate the potential interference brought about by particle turbidity, then diluted appropriately to the concentration under linear detection. The precision was the relative standard deviation (RSD) of average values of the peak areas of six parallel samples, and the accuracy was the average error of the average values of the concentration measured each day. In Table 1, the accuracy (recovery) results of DA are in the range of 95.0–105.9%, with precision (RSD) less than 3%, which demonstrates the good recovery and precision of the method we proposed. All of these consequents indicate that this approach is able to be used to analyze DA in dopamine hydrochloride injections and human urine.

**Table 1.** Intra- and inter-day accuracy (recovery) and precision (RSD) for the studied dopamine.

| Samples | Detected DA (μM) | Spiked (μM) | Within-Run [a] | | Between-Run [a] | |
|---|---|---|---|---|---|---|
| | | | Recovery (%) | RSD (%) | Recovery (%) | RSD (%) |
| Dopamine injection | 4.99 ± 0.10 | 0.80 | 95.0 | 1.08 | 96.0 | 1.51 |
| | | 4.00 | 104.2 | 1.70 | 96.7 | 2.34 |
| | | 20.00 | 96.8 | 2.03 | 105.9 | 1.58 |
| | | 70.00 (1/10)dil. | 96.5 | 0.68 | 96.0 | 0.92 |
| Urine | 56.50 ± 0.84 | 0.80 | 105.0 | 1.44 | 98.9 | 2.68 |
| | | 4.00 | 101.2 | 1.83 | 95.7 | 2.24 |
| | | 20.00 | 97.6 | 1.30 | 95.8 | 1.47 |
| | | 50.00 (1/10)dil. | 97.1 | 2.30 | 101.7 | 1.34 |

[a] n = 6.

### 3.6.3. Dilution Integrity

　　Dilution integrity was determined using spiked urine samples of dopamine hydrochloride injections and urine samples at 70 and 50 μM, respectively, and these samples were diluted with a drug-free sample as 1:10. The within- and between-run accuracy and precision were calculated as Table 1. The results show that the proposed method still has a good accuracy (<6%) and precision (<3%) that are in line with EMA guidelines. Therefore, the dilution step with the matrix had little effect on the results of the measurement.

### 3.6.4. Selectivity of the Proposed Method

　　Since many chemical components might coexist with DA in real samples, the selectivity of the proposed method was further evaluated. The effects of some interference on the fluorescence response of as-synthesized C-dots were studied by determining 12.5 μM DA with the addition of a certain amount of interference, and the results are shown in Table 2. The proposed assay strategy showed tolerance to many organic chemicals and inorganic salts such as Gly, Ser, Hcy, Cys, His, Tyr, Lys, oxalate, glucose, lactose, urea, $MnSO_4$, KCl, $MgSO_4$, $Na_2CO_3$ and $NaNO_3$. When the concentration of ascorbic acid, $ZnCl_2$ and $CaCl_2$ reached 20 times that of DA, the method still showed better selectivity. As for 5-HT, when its concentration exceeded 10 times that of DA, it showed low interference (the signal changes were less than 5%). The RSD values were less than 5%, indicating a high precision of the proposed method. It is worth noting that $FeCl_2$ and $CuCl_2$ showed a certain influence on the determination of DA. The main reason for the interference of these heavy metal ions is that under weak alkaline conditions (pH = 8.0), metal ions are hydrolyzed to cause precipitation. In addition, we also conducted interference experiments such as tartrazine, metronidazole, epinephrine and glutathione. When their concentrations were a one-time DA solution, the value of $F_0/F$ was 4.78 for tartrazine, and 1.26, 1.21, 1.06, 1.15 and 1.03 for metronidazole, epinephrine and glutathione, respectively. It can be seen that these interfering substances had a certain effect on the determination of DA, but the effect was not significant. Recently, we reported using similar methods to detect tartrazine and metronidazole [20–22]. The actual samples were food, medicine and rabbit blood. In this study, the actual samples we tested were drugs and urine. In the tartrazine and nail in urine, the content of niridazole is much lower than that of dopamine, and high concentrations of these interfering substances in pharmaceuticals are even more unlikely. Therefore, tartrazine and metronidazole did not interfere with the detection of the actual samples, and it can be concluded that the as-prepared C-dots can be used for the detection of DA in drugs and urine.

**Table 2.** Interference effect on the determination of 12.5 μM DA (n = 6).

| Interference | Con. (μM) | Signal Change (%) | RSD (%) | Interference | Con. (μM) | Signal Change (%) | RSD (%) |
|---|---|---|---|---|---|---|---|
| Gly | 2500 | 0.27 | 2.42 | $MnSO_4$ | 1250 | −4.12 | 3.00 |
| Ser | 2500 | 2.10 | 2.65 | KCl | 1250 | 0.36 | 1.56 |
| Hcy | 2500 | 4.04 | 2.45 | $MgSO_4$ | 1250 | 0.98 | 0.27 |
| Cys | 1250 | −1.27 | 1.27 | $Na_2CO_3$ | 1250 | 1.06 | 2.01 |
| His | 1250 | −1.08 | 1.78 | $NaNO_3$ | 1250 | 2.00 | 2.48 |
| Try | 1250 | −2.72 | 2.16 | Ascorbic acid | 250 | 3.72 | 3.67 |
| Lys | 1250 | 1.36 | 2.67 | $ZnCl_2$ | 250 | 2.08 | 3.05 |
| Oxalate | 1250 | −1.14 | 1.32 | $CaCl_2$ | 250 | −1.01 | 3.12 |
| Glucose | 1250 | 0.76 | 2.19 | 5-HT | 125 | −4.53 | 3.80 |
| Lactose | 1250 | 0.50 | 2.98 | $FeCl_2$ | 25 | 4.64 | 2.40 |
| Urea | 1250 | −3.14 | 2.31 | $CuCl_2$ | 25 | 4.20 | 2.92 |

### 3.6.5. Robustness

　　During the analysis run, the instrument parameters are all determined by the user. It can be seen in Figure 9 that in the fluorescence spectrum, the wavelength position at the

maximum fluorescence intensity value was substantially unchanged, and the peak shape also exhibited a normal smooth state, which was combined with the standard curve and detection result in the actual sample, indicating that the proposed method for DA detection had good robustness.

### 3.6.6. Stability

The stability of the analysis was evaluated in two ways: a working solution and an unspiked sample. Samples were placed in the refrigerator for 2 weeks at 4 °C then tested several times within a month to examine their stability. The results showed that the fluorescence performance was stable and met the methodological requirements discussed above. Since the detection condition in this work is 45 °C, the constant temperature and detection were performed 3 and 6 h after the standard sample was cooled, and the result was still stable. Therefore, we can be sure that the method we proposed has reliable stability.

### 3.7. Method Comparison

To confirm the superiority of this method, a comparison of performance was carried out between our work and other reported techniques in linear range and sensitivity. The results are presented in Table 3. In terms of both linearity and sensitivity, our C-dot sensor was comparable to the electrochemical sensor [32]. The LOD of our method was 7 to 20 times lower than in the methods using spectrophotometry [33] and another electrochemical sensor [34]. A much wider linear range (0.5–100 μM) was used in our C-dot sensor method than those using hydrogen-bonding recognition and colorimetric detection (0.2–1.10 μM) [35] and colorimetric detection (0.0–10 μM) [36]. In addition, the LOQ value in our method was 306.9 nM. Although the LOD of our method is not the lowest value in Table 3, our method could be used as an alternative to others in the determination of DA in samples. It is worth mentioning that the detection method we proposed has the advantages of simple equipment, a low cost and strong operability. Compared with other methods that require special equipment, complicated techniques or complicated processes, this method we proposed is more operable and more suitable for the routine detection of DA in clinical samples.

**Table 3.** Comparison of different methods for the determination of DA.

| Method | Linear Range (μM) | LOD (nM) | Refs. |
|---|---|---|---|
| UV-visible absorption spectroscopy | 0.84–210.90 | 632.8 | [33] |
| Pristine graphene electrochemical sensor | 5.0–710 | 2000 | [34] |
| Hydrogen-bonding recognition and colorimetric detection | 0.2–1.10 | 70 | [35] |
| Non-aggregation colorimetric sensor (based on AuNRs-Ag$^+$ system) | 0.0–10 | 200 | [36] |
| Electrochemical sensor (Ag@C/Au nanocomposites) | 0.5–4278 | 210 | [37] |
| Dual-modal (optical and electrochemical) probe | 1–10 | 410 | [38] |
| Electrochemical detection (monolayer-modified gold electrode) | 1.5–100 | 500 | [39] |
| Differential pulse voltammetry (DPV) | 0.01–1 | 24 | [40] |
| Surface-enhanced Raman spectroscopy | 0.0000001–0.001 | - | [41] |
| Triboelectric nanosensor (based on poly(tetrafluoroethylene) layer, nanoparticle arrays and aluminum film) | 10–1000 | 500 | [42] |
| Electrochemical sensor (ds-DNA with molecularly imprinted polymer) | 0.02–7 | 6 | [43] |
| Colorimetric method (Cu (II)-based metal-organic xerogels nanozyme) | 0.5–20 | 85.8 | [44] |
| Electrochemical detection (multiwalled carbon nanotube (MWCNTs)/Nafion-modified carbon tape electrode) | 0.01–1 | 10 | [45] |
| Electrochemical biosensors (Au/nanoporous stainless steel electrode) | 8.0–2000 | 70 | [46] |
| Electrochemiluminescence (GC electrode modified with CdSeTe/ZnS core−shell QDs) | 3.7–450 | 100 | [47] |
| Aptamer biosensor (aptamer complementary strand-invertase-AuNPs probe) | 0.08–100 | 30 | [48] |
| Cyclic voltammetry (carbon fiber microelectrodes) | 0–130 | 270 | [49] |
| Fluorescence probe (silicon nanoparticles) | 0.005–10.0 | 0.3 | [50] |
| Fluorescence quenching (CdS quantum dots) | 1.0–17.5 | 680 | [51] |
| Fluorescence probe (C-dots) | 0.5–100 | 93 | This work |

## 4. Conclusions

In summary, we established a method for the determination of DA based on its fluorescence quenching of C-dots. This C-dots fluorescence sensor for DA performs several outstanding functions, such as an easiness of C-dot sensor synthesis (one-pot synthesis), environmentally friendly water usage for sample preparation, a fast response, wider linearity (0.5–100 μM), outstanding selectivity and the ability to analyze many samples per day. This probe can be implemented to sensitively and selectively determine DA in human urine and dopamine hydrochloride injection clinical samples. We look forward to further research using carbon quantum dots as fluorescent probes to selectively detect drugs and dopamine in human urine in clinical practice. Further, C-dot-based visual sensor development may combine this powerful and interesting application to sensing, which could lead to a novel boulevard for sensor construction and the determination of various kinds of analytes with excellent sensitivity and selectivity.

**Supplementary Materials:** The following are available online at https://www.mdpi.com/2227-9717/9/1/170/s1, Figure S1: Fluorescence spectrum (A) and fluorescence intensity (B) of C-dots in different volumes of water when the mass of arginine was 0.1 g at a temperature of 160 °C for 4 hours ($\lambda_{em}$ = 345 nm), Figure S2: Fluorescence spectrum (A) and fluorescence intensity (B) of C-dots under various temperature when the amount of water remains unchanged at a time of 4 h ($\lambda_{em}$ = 345 nm), Figure S3: Fluorescence spectrum (A) and fluorescence intensity (B) of C-dots under various reaction time when the amount of water remains unchanged at a temperature of 180 °C ($\lambda_{em}$ = 345 nm), Figure S4: The quenching fluorescence spectra of the system by DA; Inset: the calibration graph of the corrceted fluorescence quenching and different concentrations of DA, Figure S5: The UV-vis absorption spectrum of C-dots and DA at different concentrations.

**Author Contributions:** Conceptualization, X.Y.; methodology, F.T. and H.X.; software, H.X.; validation, F.T. and S.W.; formal analysis, L.Z.; investigation, J.Z.; writing—original draft preparation, F.T. and H.X.; writing—review and editing, S.W.; supervision, X.Y. All authors have read and agreed to the published version of the manuscript.

**Funding:** This work was supported by the National Natural Science Foundation of China (21777130, 21277109), the Natural Science Foundation of Sichuan Province of China (19YYJC2801), the Meritocracy Research Funds of China West Normal University (463132) and the Fundamental Research Funds of China West Normal University (416390).

**Institutional Review Board Statement:** Not applicable.

**Informed Consent Statement:** Not applicable.

**Data Availability Statement:** The data presented in this study are available on request from the corresponding author.

**Conflicts of Interest:** The authors declare no conflict of interest.

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
