# Peer review of "Selective Determination of Dopamine in Pharmaceuticals and Human Urine Using Carbon Quantum Dots as a Fluorescent Probe"

_processes, doi:10.3390/pr9010170_

Round 1

Reviewer 1 Report

This paper by Xiupei Yang et al. constitutes an original article about carbon quantum dots used as fluorescent probes for selective determination of dopamine in pharmaceuticals and human urine. Although the manuscript seems to be interesting, in my opinion it is premature for publication with much more analysis being required – it does not merit to be published in Processes journal in the current form. Below you can find some major and minor points.

Major points:

  1. In line 216 Authors claim that: “If it is a dynamic annihilation mechanism, it conforms to the Stern-volmer equation”. However, if the fluorescence quenching is purely static, the relationship between F0/F and [Q] will be linear as well. Please explain.
  2. It seems that all experiments were performed without absorbance control. UV-Vis absorption spectra of carbon QDs should be recorded in the presence of increasing concentrations of DA and according to that potential inner filter effect should be taken into consideration.
  3. Figure 5. It is no longer sufficient to conclude that quenching is static or dynamic based on temperature measurements. Time-resolved measurements should be performed and appropriate fluorescence lifetimes should be determined in the absence and presence of dopamine. Furthermore, the determined by Authors kq values (Table 1) are two orders of magnitude higher than diffusion-controlled value in water (2x10^10). If the quenching is dynamic, kq values should be lower than the limiting value. Furthermore, it looks that kq values were determined based on assumption that a fluorescence lifetime of QD is 10 ns. On what ground? Did Authors determine that value?
  4. Once there is term “Stern-Volmer” (line 220), another time “Stern-volmer” (line 216), and then “Stern Volmer” (line 223). The units are not uniformly formatted. Once there are C degrees, another time K. Authors do not pay attention to upper and lower indexes (see for example lines 228, 279, and 338). In the text correlation coefficient is r, while in figure 7 it is R. The readers once can find that QDs were excited at 295 nm and another time that in 296 nm (line 175). All these prove that manuscript was written very carelessly and should be significantly improved.
  5. Figure 6c. It seems that in the presence of a fixed amount of DA at 25 Celsius degrees fluorescence intensity decreases successively at least for 3 hours. How is it possible if a dynamic quenching is postulated? In my opinion in case of fluorescence quenching the decrease of fluorescence intensity of a fluorophore in the presence of a quencher is instantaneous. How did Authors exclude a possibility of a reaction between QDs and DA?
  6. Line 240: what does the term “the addition of DA excitation at 345 nm” mean?
  7. Figure S2A: there is something wrong with a unit of temperature in the legend.
  8. Figures S1-S3: Emission wavelength should be added.
  9. Figure 1: the whole UV-Vis spectrum should be shown.
  10. Table 3: there is a wrong value for signal change in case of MgSO4.

Minor points:

  1. Line 31: there should be “neurotransmitters” instead of “neurotransmitter”.
  2. Line 75: there should be comma after Na2CO3.
  3. Line 97: there should be “heated” instead of “heating”.
  4. Line 114: there should be “were” instead of “was”.
  5. Line 174: there should be “spectrum” instead of “spectra” (twice).
  6. Line 221: there should be “bimolecular quenching rate constant” instead of “bimolecular quenching constant”.
  7. Line 244: there should be “decreased” instead of decreases”
  8. Line 290: there should be “linear” instead of “liner”.

Author Response

Responses to the Reviewer 1's comments and changes made

This paper by Xiupei Yang et al. constitutes an original article about carbon quantum dots used as fluorescent probes for selective determination of dopamine in pharmaceuticals and human urine. Although the manuscript seems to be interesting, in my opinion it is premature for publication with much more analysis being required – it does not merit to be published in Processes journal in the current form. Below you can find some major and minor points.

Response

Thanks for the reviewer’s precious comments. We have revised our manuscript in accordance with the reviewer’s comments. Therefore, we hope that the reviewer can accept our response and the revised manuscript.

Major points:

  1. In line 216 Authors claim that: “If it is a dynamic annihilation mechanism, it conforms to the Stern-volmer equation”. However, if the fluorescence quenching is purely static, the relationship between F0/F and [Q] will be linear as well. Please explain.

Response

Thanks for your corrections and suggestions. We have investigated the possibility of static quenching.We further explored the mechanism of DA quenching of C-dots, retested the Stern-Volmer equation, fluorescence lifetime and UV-vis absorption spectrum, and found that dynamic quenching and static quenching may exist simultaneously during the experiment. There was a certain competitive relationship between them.

Changes

Section 3.4 has been rewritten as follows.

In general, fluorescence quenching is caused by a decrease in fluorescence quantum yield due to various molecular interactions such as electron or energy transfer, molecular collision, excited state reaction, and ground state complex formation. Fluorescence quenching can be roughly divided into dynamic quenching, static quenching and a mixture of the two. Static quenching is a quenching process caused by the complex reaction of quencher and fluorescent molecules in the ground state. Dynamic quenching is a fluorescence quenching process in which a collision between a quencher and a fluorescent substance's excitation molecule loses energy and returns to the ground state without radiation. If it is a dynamic annihilation mechanism, it conforms to the Stern-Volmer equation.

F0/F = 1 + Kqτ0[Q] = 1 + KSV[Q]

in this equation F0 and F are the fluorescence intensities of C-dots at 347 nm in the absence and presence of DA, respectively; KSV and Kq are the Stern−Volmer quenching constant and the biolecular quenching rate constant t, respectively; [Q] is the concentration of DA and τ0 is the average lifetime of C-dots in the absence of DA. Then we explored the quenching mechanism of DA on C-dots. Fig. 5 shows the fluorescence intensities of the C-dots analyzed by plotting F0/F versus [Q] at 25, 35, 45, and 55℃, respectively. Table 1 shows the Stern-Volmer quenching constant KSV of DA for C-Dots at different temperatures. This indicates that the possible quenching mechanism of DA on C-Dots fluorescence is a dynamic quenching process, because the law of KSV increasing with increasing temperature conforms to the Stern-Volmer equation. At the same time, we tested the fluorescence lifetime of carbon quantum dots (Fig. S4). After DA was added, the fluorescence lifetime remains unchanged and its value remains 3.55 ns, and the ultraviolet absorption spectrum (Fig. S5) had a weak response absorption peak of a new substance, and the peak intensity increased with the increased of DA concentration. This indicates that the static quenching mechanism may exist. Therefore, we speculate that there are two mechanisms of dynamic quenching and static quenching in the quenching mechanism of DA on C-dots, and there is a competitive relationship between the two mechanisms.

Figure 5. Stern−Volmer plots for the system of C-dots−dopamine under temperatures of 25, 35, 45, and 55℃, respectively. F0 and F are the fluorescence intensity of C-dots in the absence and presence of dopamine, respectively. Conditions: C-dots, 800 μL; PB, 30 mM, pH8.0.

Table 1. Stern–Volmer quenching constants for the interaction of C-Dots and DA at different temperatures.

pH

T()

KSV(L·mol−1)

Kq(L·mol−1·s−1)

r2

SD

8.0

25℃

2.211×104

6.227×1012

0.9680

0.0301

8.0

35℃

2.287×104

6.441×1012

0.9942

0.0265

8.0

45℃

2.387×104

6.708×1012

0.9924

0.0134

8.0

55℃

3.056×104

8.610×1012

0.9927

0.0253

  1. It seems that all experiments were performed without absorbance control. UV-Vis absorption spectra of carbon QDs should be recorded in the presence of increasing concentrations of DA and according to that potential inner filter effect should be taken into consideration.

Response

Thanks for your corrections and suggestions. We have added the UV-vis absorption spectra of the blank group containing only carbon quantum dots and DA solutions with different concentrations, as shown in Figure S5. The UV spectrum shows that by adding DA, the position of the UV absorption peak of the carbon quantum dot solution does not change, but a new characteristic absorption peak appears. Combined with the fluorescence lifetime (Figure S4), we speculate that there may be a static quenching mechanism, which may be related to the internal rate effect. Therefore, in the quenching mechanism, we speculate that dynamic quenching and static quenching may be both, but there was a certain competitive relationship between them.

Changes

At the same time, we tested the fluorescence lifetime of carbon quantum dots (Fig. S4). After DA was added, the fluorescence lifetime remains unchanged and its value remains 3.55 ns, and the ultraviolet absorption spectrum (Fig. S5) had a weak response absorption peak of a new substance, and the peak intensity increased with the increased of DA concentration. This indicates that the static quenching mechanism may exist. Therefore, we speculate that there are two mechanisms of dynamic quenching and static quenching in the quenching mechanism of DA on C-dots, and there is a competitive relationship between the two mechanisms.

Fig. S5 The UV-vis absorption spectrum of C-dots and DA at different concentrations.

  1. Figure 5. It is no longer sufficient to conclude that quenching is static or dynamic based on temperature measurements. Time-resolved measurements should be performed and appropriate fluorescence lifetimes should be determined in the absence and presence of dopamine. Furthermore, the determined by Authors kq values (Table 1) are two orders of magnitude higher than diffusion-controlled value in water (2x10^10). If the quenching is dynamic, kq values should be lower than the limiting value. Furthermore, it looks that kq values were determined based on assumption that a fluorescence lifetime of QD is 10 ns. On what ground? Did Authors determine that value?

Response

Thanks for your suggestions. We have already measured the fluorescence lifetime (Fig. S4), the basically stable lifetime value as 3.55 ns and the new peaks appearing in the ultraviolet indicate the possibility of static quenching. However, the Stern-volmer equation gradually increases with the temperature, and the trend of the slope Ksv gradually increases indicates the possibility of dynamic quenching. Therefore, we speculate that dopamine quenching C-dots may be dynamic quenching and static quenching at the same time, and there is a competitive relationship.

Changes

At the same time, we tested the fluorescence lifetime of carbon quantum dots (Fig. S4). After DA was added, the fluorescence lifetime remains unchanged and its value remains 3.55 ns, and the ultraviolet absorption spectrum (Fig. S5) had a weak response absorption peak of a new substance, and the peak intensity increased with the increased of DA concentration. This indicates that the static quenching mechanism may exist. Therefore, we speculate that there are two mechanisms of dynamic quenching and static quenching in the quenching mechanism of DA on C-dots, and there is a competitive relationship between the two mechanisms.

Fig. S4 The fluorescence lifetime of C-dots in the presence/absence of DA.

  1. Once there is term “Stern-Volmer” (line 220), another time “Stern-volmer” (line 216), and then “Stern Volmer” (line 223). The units are not uniformly formatted. Once there are C degrees, another time K. Authors do not pay attention to upper and lower indexes (see for example lines 228, 279, and 338). In the text correlation coefficient is r, while in figure 7 it is R. The readers once can find that QDs were excited at 295 nm and another time that in 296 nm (line 175). All these prove that manuscript was written very carelessly and should be significantly improved.

Response and changes

I am very sorry for our carelessness. We have changed both “Stern-volmer” and “Stern Volmer” to “Stern-Volmer ”. And changed “F0/F” to “F0/F”. Besides, we have united the temperature unit of Celsius in our article, changed “ Fig. 5 shows the fluorescence intensities of the C-dots analyzed by plotting F0/F versus [Q] at 298, 308, 318, and 328 K, respectively.” to “Fig. 5 shows the fluorescence intensities of the C-dots analyzed by plotting F0/F versus [Q] at 25, 35, 45, and 55 ℃, respectively.”

  1. Figure 6c. It seems that in the presence of a fixed amount of DA at 25 Celsius degrees fluorescence intensity decreases successively at least for 3 hours. How is it possible if a dynamic quenching is postulated? In my opinion in case of fluorescence quenching the decrease of fluorescence intensity of a fluorophore in the presence of a quencher is instantaneous. How did Authors exclude a possibility of a reaction between QDs and DA?

Response

Thanks for your suggestions, we have made adjustments accordingly.

Changes

We measured the UV-vis absorption spectrum in the presence of different concentrations of DA  (Fig S5). The new absorption peaks in the UV-vis absorption spectrum indicated new substances may be exist, and have an internal filtration effect.

  1. Line 240: what does the term “the addition of DA excitation at 345 nm” mean?

Response and changes

Thank you for your comments. The sentence “In this work, the efficiency of fluorescence quenching is expressed by F0/F, where F and F0 are the fluorescence intensity of the C-dots solution after and before the addition of DA excitation at 345 nm, respectively.” has been rewritten as “In this work, the efficiency of fluorescence quenching is expressed by F0/F, where F and F0 are the fluorescence intensity of the C-dots solution after and before the addition of DA at 345 nm, respectively.”.

  1. Figure S2A: there is something wrong with a unit of temperature in the legend.

Response and changes

Thanks for the reviewer’s precious comments and suggestions. A unit of temperature has been revised in the legend of Figure S2A.

  1. Figures S1-S3: Emission wavelength should be added.

Response and changes

Thanks for the reviewer’s precious comments. Emission wavelength has been added in Figure S1-S3, respectively.

  1. Figure 1: the whole UV-Vis spectrum should be shown.

Response

Thank you for your suggestion.,we have supplemented the UV-Vis spectrum at 200-800 nm.

Changes

Figure 1. UV-vis absorption spectrum (a) and fluorescence emission spectrum (b) of the C-dots (excitated at 295 nm).

  1. Table 3: there is a wrong value for signal change in case of MgSO4.

Response and changes

We are very sorry for our negligence. We have rewritten the signal change value "1250" of MgSO4 to "0.98" in Table 3.

Minor points:

  1. Line 31: there should be “neurotransmitters” instead of “neurotransmitter”.

Response and changes

We are very sorry for our negligence of writing. We have modified “neurotransmitter” to “neurotransmitters”.

  1. Line 75: there should be comma after Na2CO3.

Response and changes

We are very sorry for our negligence of writing. We have corrected the content as “Na2CO3,”.

  1. Line 97: there should be “heated” instead of “heating”.

Response and changes

Thanks for your suggestion. We have made the corresponding changed to “heated”.

  1. Line 114: there should be “were” instead of “was”.

Response and changes

We are very sorry for our negligence of witting. We have made the corresponding changed “was” to “were”.

  1. Line 174: there should be “spectrum” instead of “spectra” (twice).

Response and changes

We have made correction according to the Reviewer’s comments and amended “spectra” to “spectrum” (twice).

  1. Line 221: there should be “bimolecular quenching rate constant” instead of “bimolecular quenching constant”.

Response and changes

We are very sorry for our negligence of witting. We have made the corresponding changed “bimolecular quenching constant” to “biolecular quenching rate constant”.

  1. Line 244: there should be “decreased” instead of decreases”

Response and changes

We have made correction according to the Reviewer’s comments and amended it to “decreased”.

  1. Line 290: there should be “linear” instead of “liner”.

Response and changes

We apologize for our negligence. We have modified to “linear” in the manuscript.

Reviewer 2 Report

In my opinion, the manuscript entitled ,,Selective determination of dopamine in pharmaceuticals and human urine using carbon quantum dots as a fluorescent probe’’ described by Xiupei Yang , Fangming Tian, ShaohuaWen, Hua Xu, Lin Zhang and Jie Zeng can be recommended for publication in Processes, however after minor revision.

In this study, the authors have developed the fluorescent C-dots' synthesis by L-arginine's hydrothermal treatment. Next, their analytical application for determination of dopamine in injections and urine samples has been examined. The application of carbon quantum dots as a fluorescent probe seems to be an interesting alternative to other analytical methods. Despite some limitations of the presented study, the findings seem to be interesting for publication.

My remarks and recommendations are as follows:

  1. Subsection ,,Chemical reagents'': What does it mean that dopamine hydrochloride injection samples were collected in School Hospital of China West Normal University? Has the finished dosage form of pharmaceutical formulation been used? If so, what was the name of the formulation and the manufacturer? What was the dopamine concentration in the formulation?
  2. I suggest adding "Method validation" subsection in section 2. The authors wrote in the manuscript that the validation was performed according to the guideline on bioanalytical method validation published by the European Medicines Agency (EMA). However, for the sake of clarity, more information should be provided. Please provide accurate information about calculation method of individual validation parameters, e.g. linearity, LOD, LOQ, accuracy, precision, selectivity. Please also provide appropriate references to the above-mentioned guideline.
  3. How was the standard curve prepared?
  4. What is the limit of quantification (LOQ)?
  5. The authors tested many organic chemicals and inorganic salts for selectivity evaluation. This is the right approach. According to quideline provided by the European Medical Agency the selectivity should be proved using at least 6 individual sources of the matrix. Did the authors only use urine obtained from two volunteers? The urine of different people can be chemically vary depending on different physiological and medical conditions, food intake and medications. What can be the influence of matrix? What will be the effect of other medications used that may be present in the patient's urine? Authors conducted interference experiments using only a few  compounds such metronidazole, epinephrine, and glutathione.  If there are limitations of the method, it should be clearly indicated.
  6. According to table 3 the interference effect of MgSO4 results in a signal change of 1250%. Is it correct?
  7. The authors wrote that the method is suitable for the determination of dopamine in human urine for clinical application. However, the method has not been clinically applied to an adequate number of patients. The method is interesting and promising, however, after the validation stage, its clinical usefulness should be confirmed.
  8. There are some typos in the text. The text of manuscript should be carefully checked.

Author Response

Responses to the Reviewer 2's comments and changes made

In my opinion, the manuscript entitled ,,Selective determination of dopamine in pharmaceuticals and human urine using carbon quantum dots as a fluorescent probe’’ described by Xiupei Yang , Fangming Tian, ShaohuaWen, Hua Xu, Lin Zhang and Jie Zeng can be recommended for publication in Processes, however after minor revision.

In this study, the authors have developed the fluorescent C-dots' synthesis by L-arginine's hydrothermal treatment. Next, their analytical application for determination of dopamine in injections and urine samples has been examined. The application of carbon quantum dots as a fluorescent probe seems to be an interesting alternative to other analytical methods. Despite some limitations of the presented study, the findings seem to be interesting for publication.

Response

Thanks for the reviewer's positive comments. We have thoroughly revised the manuscript according to the comments.

My remarks and recommendations are as follows:

  1. Subsection ,,Chemical reagents'': What does it mean that dopamine hydrochloride injection samples were collected in School Hospital of China West Normal University? Has the finished dosage form of pharmaceutical formulation been used? If so, what was the name of the formulation and the manufacturer? What was the dopamine concentration in the formulation?

Response

Thanks for your suggestions.

Changes

We have completed the relevant information. We have changed “Human urine and dopamine hydrochloride injection samples were collected in School Hospital of China West Normal University.” to “Dilute dopamine hydrochloride injection (10mg/L, Yuanda Pharmaceutical China Co., Ltd.) and Human urine was collected from the school hospital.” in “2.1. Chemical reagents”.

  1. I suggest adding "Method validation" subsection in section 2. The authors wrote in the manuscript that the validation was performed according to the guideline on bioanalytical method validation published by the European Medicines Agency (EMA). However, for the sake of clarity, more information should be provided. Please provide accurate information about calculation method of individual validation parameters, e.g. linearity, LOD, LOQ, accuracy, precision, selectivity. Please also provide appropriate references to the above-mentioned guideline.

Response

Thank you for your suggestion.We have added the calculation of LOQ in the article. In addition, the article has been compared with other detection methods, and the results are presented in the Table 4.

Changes

We have added the calculation of LOQ in the article as “In addition, the LOQ value in our method was 306.9 nM” in “3.7. Method comparison”. In addition, the article has been compared with other detection methods, and the results are presented in the Table 4.

  1. How was the standard curve prepared?

Response

Thanks for your question.

Changes

The process of the standard curve is as follows: In the phosphate buffer solution of pH=8, added 800 μM of C-dots solution and different volumes of dopamine hydrochloride solution, and diluted to 4 mL with high-purity water, so that the final concentration was 0.00, 0.50, 0.75, 2.50, 7.50, 12.50, 17.50, 25.00, 32.50, 40.00, 50.00, 60.00, 75.00, 100.00 μM, respectively. Then equilibrated for 1 h at 45℃.

  1. What is the limit of quantification (LOQ)?

Response

Thanks for your question. We have added LOQ value in our manuscript.

Changes

We have added the calculation of LOQ in the article as “In addition, the LOQ value in our method was 306.9 nM” in “3.7. Method comparison”.

  1. The authors tested many organic chemicals and inorganic salts for selectivity evaluation. This is the right approach. According to quideline provided by the European Medical Agency the selectivity should be proved using at least 6 individual sources of the matrix. Did the authors only use urine obtained from two volunteers? The urine of different people can be chemically vary depending on different physiological and medical conditions, food intake and medications. What can be the influence of matrix? What will be the effect of other medications used that may be present in the patient's urine? Authors conducted interference experiments using only a few  compounds such metronidazole, epinephrine, and glutathione.  If there are limitations of the method, it should be clearly indicated.

Response

Thank you for your suggestions on our work. We referred to other people's research and selected the urine of two volunteers for testing. In order to reduce the experimental error, each sample set 6 parallel samples.

Changes

We have changed “Overnight urine samples were obtained from two healthful volunteers” to “Overnight urine samples were obtained from two healthful volunteers, and six parallel samples were tested in ” 2.4. Preparation of samples”.

We inspected 22 interfering substances for detection, including amino acids, sugars, salts etc in Table 3.

  1. According to table 3 the interference effect of MgSO4 results in a signal change of 1250%. Is it correct?

Response

We are very sorry for our negligence, we have rewritted the signal change.

Changes

We have rewritten the signal change value "1250" of MgSO4 to "0.98" in the Table 3 at “3.6.4. Selectivity of the proposed method”.

  1. The authors wrote that the method is suitable for the determination of dopamine in human urine for clinical application. However, the method has not been clinically applied to an adequate number of patients. The method is interesting and promising, however, after the validation stage, its clinical usefulness should be confirmed.

Response

Thank you for your support and suggestions for our work. Based on the high sensitivity and selectivity of the probe we prepared for DA, and through experimental verification, we have proposed that it can be used in the detection of clinical samples of human urine and dopamine hydrochloride injection, which provides a new reference for clinical detection methods.

Changes

We have changed “ This probe can be implemented to sensitively and selectively determine of DA in human urine and dopamine hydrochloride injection clinical samples. Further, C-dots-based visual sensors' development may combine this powerful and interesting attribute into the sensor purpose, which could unfold a novel boulevard for sensors' construction for the determination of various kinds of analytes with excellent sensitivity and selectivity.” to “ This probe can be implemented to sensitively and selectively determine of DA in human urine and dopamine hydrochloride injection clinical samples. We also look forward to further research on using carbon quantum dots as fluorescent probes to selectively detect drugs and dopamine in human urine in clinical practice. Further, C-dots-based visual sensors' development may combine this powerful and interesting attribute into the sensor purpose, which could unfold a novel boulevard for sensors' construction for the determination of various kinds of analytes with excellent sensitivity and selectivity.” in the Section of 4 Conclusion.

  1. There are some typos in the text. The text of manuscript should be carefully checked.

Response

We are very sorry for our incorrect writing, we have corrected the wrong writing in the article.

Changes

We have changed "we have chosen"to "we we chosen",“Stern-volmer” (line 216) and “Stern Volmer” to “ Stern-Volme”, “F0/F”( lines 228, 279, and 338 ) to “F0/F”.

Reviewer 3 Report

In the Section 3.1 it is mentioned that Authors used the C-doth without further purification (without dialysis) for the tests with dopamine, but did not analyze the solution thoroughly, e.g. mentioned the presence of low molecular weight compounds from L-arginine hydrothermal treatment, but the exact composition in unknown. Such analysis should be performed.

In the Figure 2 (TEM image) the scale should be corrected due to inappropriate visability. If the presented scale is 100 nm, the average diameter of C-dots will be above 10 nm - the statistical analysis should be presented.

In the FTIR analysis the bands from carboxylic groups are not indicated appropriately, e.g. around 1415, 1240, 940 cm-1 are missing. Moreover free amino groups are not pointed, but mentioned in XPS analysis and in the Scheme 1.

In the line 216 should be "Stern-Volmer" not "Stern-volmer".

In the line 223 should be "Stern-Volmer" not "Stern Volmer".

In the line 274 should be "we have chosen" not "we we chosen".

Author Response

Responses to the Reviewer 3's comments and changes made

In the Section 3.1 it is mentioned that Authors used the C-doth without further purification (without dialysis) for the tests with dopamine, but did not analyze the solution thoroughly, e.g. mentioned the presence of low molecular weight compounds from L-arginine hydrothermal treatment, but the exact composition in unknown. Such analysis should be performed.

Response

Thanks for the reviewer's positive comments. We have thoroughly revised the manuscript according to the comments.

  1. In the Figure 2 (TEM image) the scale should be corrected due to inappropriate visability. If the presented scale is 100 nm, the average diameter of C-dots will be above 10 nm - the statistical analysis should be presented.

Response

Thank you for your suggestion on our work. In the TEM image, we can see that the size of the carbon quantum dots is not uniform, and our analysis of its particle size was also based on the average particle size proposed by statistics.

Changes

We have changed “It reveals that the C-dots are well dispersed in solution with spherical shape and have an average size of 9 nm approximately.” to “It found that the C-dots are spherical and non-uniform in particle size, with an average particle size of about 9 nm, and they are well dispersed in the solution.” in the “3.2. Characterization of C-dots”.

  1. In the FTIR analysis the bands from carboxylic groups are not indicated appropriately, e.g. around 1415, 1240, 940 cm-1 are missing. Moreover free amino groups are not pointed, but mentioned in XPS analysis and in the Scheme 1.

Response

Thank you for your suggestions, we have added the content of this part. It can be seen from the FTIR that the vibrational peak of the amino group in arginine is not significant in the carbon quantum dots, but the broadening of the vibration peaks at 3500-3300 cm-1 cm and 1650-1550 cm-1 can prove the existence of amino groups. The insignificant shape may be due to the impact of the vibration peak of the carboxyl group.

Changes

We have added “In addition, the broadening of peaks at 3500~3300 cm−1 and 1650~1550 cm−1 indicate the existence of N-H stretching vibration peaks. The reason for the insignificant peak shape may be affected by the carboxyl vibration peak.” at “ 3.2. Characterization of C-dots” in our manuscript.

  1. In the line 216 should be "Stern-Volmer" not "Stern-volmer".

Response and changes

We are very sorry for our incorrect writing. We have made the corresponding changed to “Stern-Volmer”.

  1. In the line 223 should be "Stern-Volmer" not "Stern Volmer".

Response and changes

We are very sorry for our negligence of wtitting. We have made the corresponding changed to “Stern-Volmer”. 

  1. In the line 274 should be "we have chosen" not "we we chosen".

Response and changes

We have made correction according to the Reviewer’s comments, and made the corresponding changes to “we have chosen”.

Round 2

Reviewer 1 Report

The authors have addressed most of the comments made. Nevertheless, there are still concerns about the mechanism involved and consequently this manuscript still needs significant improvement.

  1. Why the old Figure 5 is different than new Figure 5? Why the plots have been replaced? It seems that both experiments were performed under the same conditions. In that case why the new values of Stern-Volmer quenching constants differ from the old ones? What is the difference between two versions?
  2. Authors claim that “the ultraviolet absorption spectrum (Fig. S5) had a weak response absorption peak of a new substance, and the peak intensity increased with the increased of DA concentration”. However to state about a change in UV-Vis spectrum, appropriate spectra of mixtures of CDs and DA should be compared with the ones of pure DA at various concentrations and CDs. The fact that absorption increases does not prove the interaction and consequently static quenching – it can be just the effect of the addition of DA which absorbs strongly. Authors “speculate that there are two mechanisms of dynamic quenching and static quenching in the quenching mechanism of DA on C-dots”. But on what ground? Fluorescence lifetime is fixed in the absence and presence of a quencher. Furthermore, if there is a combined quenching why Stern-Volmer plots are strictly linear instead of positive deviation towards Y axis? Then Authors claim that “there is a competitive relationship between the two mechanisms”? But what relationship? On what ground? Each statement should be confirmed with appropriate reference. Finally, I still do not understand how is it possible that fluorescence intensity (assuming quenching) in the presence of a fixed amount of DA does not decrease instantaneously - please cite appropriate reference to confirm it is reported.
  3. It seems that fluorescence intensity values still were not corrected for inner filter effect. Since dopamine absorbs strongly at excitation and emission wavelengths of CDs it is essential. The fluorescence intensity values should be corrected according to a formula Fcorr = Fobs*10^[(Aex+Aem)/2]. See for example Zamojc, K., Wiczk, W., & Chmurzynski, L. (2018). The influence of the type of substituents and the solvent on the interactions between different coumarins and selected TEMPO analogues–Fluorescence quenching studies. Chemical Physics, 513, 188-194.

Author Response

Thank you for your comments. We have thoroughly revised the manuscript according to the comments. Attached please find the Responses to the Reviewer 1's comments and changes made.

Reviewer 3 Report

All the suggestions were applied and the article should be published without changes.

Author Response

Thanks